# Accurate interpretation of within-host dissemination using barcoded bacteria

Rachel T. Giorgio,[1] My T. Le,[1] Ting Zhang,[2] Caitlyn L. Holmes,[3] Karthik Hullahalli[1]

**ABSTRACT**  Bacterial dissemination across tissues is a critically important process influencing infection outcomes. Monitoring within-host dissemination is challenging because conventional measures of bulk bacterial burden cannot distinguish between lineages that are shared between tissues and those that replicate locally. This limitation can be overcome using barcoded bacteria, where deep sequencing of the barcode locus and comparisons of barcodes between tissues define which lineages spread within the host. Numerous studies have used barcoded bacteria to generate high-resolution maps of dissemination. However, since multiple cells in the infectious inoculum can contain identical barcodes, inferences about dissemination can be confounded when distinct lineages from the inoculum with identical barcodes are observed in different tissues. Thus, even though the same barcodes can be observed in different tissues, dissemination between these tissues may not have occurred. Here, we aimed to develop an approach that would provide a solution to this confounding effect. We developed a simulation-based distance metric that quantifies the significance of observing shared barcodes between tissues. We validated this approach using simulated data sets spanning three orders of magnitude in barcode diversities and on three published experimental infection data sets. Our reanalysis reveals previously unappreciated patterns of *Escherichia coli* spread during liver abscess formation, clarifies the role of the Muc2 mucin in *Listeria monocytogenes* systemic spread, and quantifies how *Klebsiella pneumoniae* replication in the lungs drives systemic dissemination. As barcoding studies expand across diverse infection models, this approach provides an essential tool for accurate interpretation of within-host bacterial dissemination.

**IMPORTANCE**  How microbes move between tissues in the host is an important factor that controls the outcome and severity of infections. A powerful method to monitor within-host microbial dissemination is the use of barcoded bacteria and lineage tracing. Comparisons of barcodes between tissues enable inferences of microbial dissemination, and this method has been applied to diverse contexts of bacterial infections. Here, we demonstrate that inferences of microbial dissemination are confounded, where observing identical barcodes in different tissues does not always signify that dissemination has occurred. To overcome this limitation, we define a metric to quantify the extent to which sharing of barcodes is meaningful and provide new insights into previous barcoding studies in *Escherichia coli, Listeria monocytogenes*, and *Klebsiella pneumoniae*. As bacterial lineage tracing continues to be applied across diverse models, our method will help ensure accurate interpretations of microbial dissemination.

**KEYWORDS**  lineage tracing, barcodes, dissemination

Address correspondence to Karthik Hullahalli, khullahalli@luc.edu.

The authors declare no conflict of interest.

The severity of bacterial infections is governed by several factors, such as the virulence of the pathogen, the immune status of the host, and the site of initial infection. The interplay of these variables during infection leads to numerically distinct levels

of microbes across tissues, where high bacterial burdens are typically associated with worse outcomes. In addition to the number of bacteria within a tissue, a major factor that governs the severity of infection is the extent to which bacterial pathogens can disseminate between tissues. For example, high bacterial burdens in a tissue may be effectively restricted and unable to migrate to other sites in the body, leading to overall positive clinical outcomes (1). In contrast, lower bacterial burdens in a tissue may have severe consequences for the host if these bacteria are able to escape to secondary sites (2). Thus, in order to understand the processes that control infection severity, it is essential to decipher how pathogens disseminate across tissues.

Traditional methods to quantify microbial dissemination typically leverage the same variable as methods to quantify bacterial burden: bulk measurements of total colony-forming units (CFUs) in a tissue. Historically, dissemination is inferred based on which tissue is being measured. If CFU burdens are high in a tissue that is distal to where pathogens were inoculated, it is reasonable to infer that dissemination has occurred between the inoculation site and the distal tissue (3). However, inferences of dissemination made from bulk CFU measurements are limited because they fail to directly measure how the microbial populations at the distal site arose. For example, the majority of the population at the distal site may have actually arisen from *in situ* replication following translocation of only a small number of microbes. In contrast, there may be an alternative tissue that was not measured but is the true reservoir for the bacterial population at the distal site. Bulk CFU measurements also obfuscate important aspects regarding the dynamics of dissemination, such as whether translocation occurs in a single step or is continuous, or whether a small or large number of clones are migrating. Thus, bulk measurements of CFU are not capable of accurately quantifying microbial dissemination.

The inability of CFU measurements to accurately quantify dissemination results from the fact that the total bacterial burden is unable to decipher which lineages are spreading between tissues and which are replicating locally (4). Overcoming this limitation is possible through the introduction of genetic diversity in the form of short, random nucleotide sequences integrated into a conserved fitness-neutral site in the genome, commonly known as barcodes (5). Through PCR and deep sequencing of the barcodes in a population, it becomes possible to infer which clones from the inoculum gave rise to the population at various sites of infection, known as the founding population. The observation that barcodes (and thus presumably founders) between two tissues are shared would indicate that these two tissues share a common pool of replicative microbes, which can then be interpreted against the anatomical context to infer which tissue may be the upstream reservoir. In contrast, the observation that the barcodes between two tissues are distinct would indicate that replication largely occurs *in situ* and that these two sites do not exchange microbial populations. Over the past decade, a growing body of literature has leveraged barcoded bacteria to decipher the paths of microbial dissemination across a wide range of infection contexts (6–21). These studies have markedly heightened the resolution by which we can understand the dynamics of bacterial infections.

The idealized barcoded bacterial library is one that contains an infinite number of barcodes, such that every cell in the inoculum has a unique barcode (Fig. 1A). In this ideal scenario, observing the same barcode in two tissues necessarily indicates that this clone has replicated and a subpopulation (containing identical barcodes) has spread between the two tissues. This logic is necessary since, in this idealized barcode library, each barcode only exists in one cell in the inoculum and cannot exist in two sites simultaneously without replication and dissemination. However, it is impractical to create a library with an infinite number of barcodes, and all barcoding studies have used libraries of finite diversities, from a few hundred to tens of thousands of barcodes (9, 22–24). A consequence of having a finite number of barcodes is that it is possible that two tissues share identical barcodes simply due to independent seeding of these two tissues by different founders possessing the same barcode (Fig. 1B). With small founding population sizes relative to the number of barcodes in the library, this issue is negligible;

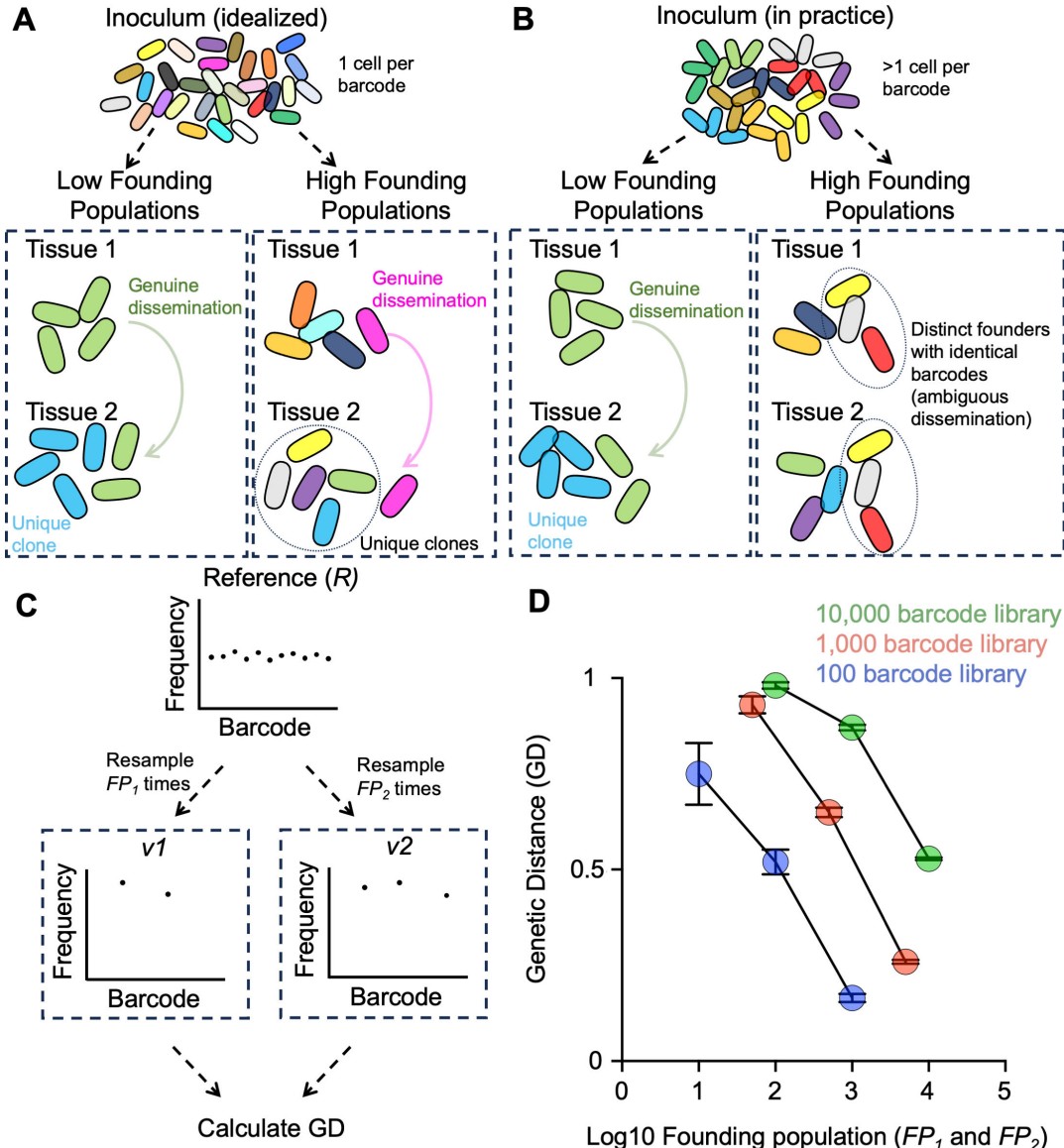

**FIG 1** Higher founding population sizes yield lower genetic distance (GD) values. (A) The idealized barcoded library contains an infinite number of barcodes (schematized by different colors), where every cell in the inoculum possesses a unique tag. Regardless of founding population sizes, the observation that the same barcode is observed in different tissues necessitates that the clone has replicated in and disseminated across compartments. (B) A diverse inoculum containing a large but finite number of barcodes can give rise to populations with low or high founding population sizes. With low founding population sizes, the observation that two tissues share identical barcodes likely indicates that the clones replicated and disseminated between the tissues. In contrast, at higher founding population sizes, it is possible that observing identical barcodes in two tissues is due to independent sampling of distinct founders that possess identical barcodes (ambiguous dissemination). (C) Simulations can be used to illustrate the influence of founding population sizes on GD values. A reference barcode library ($R$) can be resampled across various founding population sizes ($FP_1$ and $FP_2$), and the GD between these computational samples can be calculated. (D) Using the simulation procedure in panel C, GD is calculated across various founding population sizes. Across a range of barcode library diversities, higher FPs yield lower GDs. Error bars represent means and standard deviations.

for example, in a library containing 10,000 unique barcodes, it is highly unlikely that two tissues that possess the same three barcodes acquired them from distinct founders. However, with smaller libraries or with higher founding population sizes, it is more likely that tissues share barcodes in a manner purely due to random sampling of the same barcodes across different founders. Thus, interpretations of dissemination from barcoding experiments are highly confounded by founding population sizes.

Here, we present a solution to the problem of finite barcoded libraries confounding interpretations of dissemination. We create a distance metric that compares the similarity of barcodes observed in biological samples to the similarity of barcodes obtained by simulations of samples with equal founding population sizes. This distance metric is used to interpret whether the similarity of barcodes between two samples is greater than would be expected by random chance. Using simulations, we show that our approach is robust across a 1,000-fold range in founding population sizes and a 100-fold range in library diversities. We then apply this method to three published barcoding data sets to uncover new patterns of dissemination. In a model of *Escherichia coli* liver abscess formation, our reanalysis demonstrates that abscesses are minor but significant sources of microbial dissemination. In a model of *Listeria* systemic infection, our analysis identifies confounding effects of high founding population sizes and reframes the role of the Muc2 mucin in microbial dissemination. Finally, in a model of *Klebsiella* bacteremic pneumonia, our reanalysis directly quantifies the extent to which dissemination requires upstream replication and how dissemination from the lung changes over time. Together, our results present a solution to a major challenge in bacterial barcoding experiments, and we anticipate that this approach will be a powerful analytic method in the microbial lineage tracing toolbox.

## RESULTS AND DISCUSSION

### Interpretations of dissemination by genetic distances are confounded by founding population sizes

Barcoding experiments have leveraged a genetic distance (GD) metric to infer whether populations of microbes are being exchanged between two compartments. When the barcode distributions between two tissues are similar, GD values are low; when barcode distributions are dissimilar, GD values are high. A decrease in GD values is often interpreted as increased dissemination between compartments, which is reasonable given that the decrease in GD indicates that barcode distributions are more similar and potentially indicates increased sharing of clones. However, if the decrease in GD values is accompanied by a corresponding increase in founding population sizes, such interpretations may be confounded. Since libraries consist of a finite number of unique barcodes (i.e., the same barcode can be represented by multiple cells in the inoculum), an increase in founding population sizes may result in greater barcode similarity. Higher founding population sizes increase the likelihood that the same barcode will be independently represented by different founders across multiple tissue sites (Fig. 1B).

We illustrate the ability of founding population sizes to confound interpretations of GDs through simulations (Fig. 1C). We define a reference barcode frequency vector $R$ that consists of $n$ elements representing the frequency of individual barcodes $b_1, b_2, b_3 \ldots b_n$. In biological samples, $n$ reflects the number of unique barcodes in the library, and the sum of all $n$ elements in $R$ is equal to 1. $R$ can be multinomially resampled $FP_1$ (founding population size of sample 1) times and scaled such that the sum of all elements remains 1, yielding a new barcode frequency vector $v_1$. Biologically, this resampling simulates a single-step bottleneck, and $FP_1$ represents the founding population size at a given site. We can similarly simulate a bottleneck at a different site, yielding a second barcode frequency vector $v_2$. We define the GD between $v_1$ and $v_2$ as

$$\text{GD}_{v1,v2} = \sqrt{1 - \sum_{i=1}^{n} \sqrt{(b_i, v_1)(b_i, v_2)}},$$

where $b_i, v_1$ represents the frequency of the $i$th barcode in $v_1$, and $b_i, v_2$ represents the frequency of the $i$th barcode in $v_2$. Note that our previous studies multiply the resulting values by $(2\sqrt{2})/\pi \sim 0.9003$ (4, 25, 26). Here, we omit this constant for clarity, so that GD values can range from 0 to 1, rather than from 0 to 0.9003.

To demonstrate the impact of varying founding population sizes on GD, we vary $FP_1$ and $FP_2$ across a range of values and library diversities and calculate the resulting GD between $v_1$ and $v_2$ (Fig. 1D). Biologically, this procedure simulates bottlenecks at varying sizes across two tissue sites. Importantly, since both populations are sampled independently, there is, by definition, no dissemination across these sites. However, when plotting GD values across varying founding population sizes, it is evident that higher founding population sizes, relative to the total number of barcodes in the library, yield smaller GD values, despite the fact that there is no dissemination between the simulated compartments. Thus, GD values can be low due to relatively high founding population sizes, rather than genuine dissemination between compartments, a phenomenon we refer to as "ambiguous" dissemination for the remainder of this manuscript. Importantly, numerous studies have reported contexts with reduced GD values, some of which coincide with increased founding population sizes, necessitating a metric to evaluate the significance of GDs (9, 24, 27–29).

## A simulation-based approach to quantify the significance of GD values

We reasoned that, if ambiguous dissemination could be modeled via multinomial resampling-based simulations, these same simulations could be used as comparators to measure the significance of biologically observed GD values. For example, if $GD_{v1,v2}$ is 0.5, but simulations of two random samples with equivalent founding population sizes as $v_1$ and $v_2$ consistently yield higher GD values, we can conclude that the GD value is lower than would be expected by random chance and that the sharing of barcodes between the two sites reflects genuine dissemination. In contrast, if simulations yield GD values close to 0.5, dissemination between these two sites is indistinguishable from random chance and likely ambiguous.

We define a metric to measure the extent to which GD values are expected based on random chance as the DREX score (distance from randomly sampled expectations, Fig. 2). To define the DREX score between $v_1$ and $v_2$, which represent barcode frequency vectors at tissue 1 and tissue 2, respectively, we first obtained founding population sizes $FP_1$ and $FP_2$ as outputs of the STAMPR pipeline (25). STAMPR calculates founding population sizes through a metric known as Ns, which represents sampling depth (from multinomial resampling of $R$) required to observe the given number of barcodes in a sample, irrespective of their relative abundances. Using the same reference barcode vector as was used for biological samples ($R$), we performed multinomial resampling with founding population sizes $FP_1$ and $FP_2$ to generate two simulated barcode frequency vectors, $p$ and $q$. $GD_{p,q}$ is calculated, and this simulation procedure is repeated 50 times, generating a distribution of simulated $GD_{p,q}$ values. We define the DREX score of $v_1$ and $v_2$ as the z-score of $GD_{v1,v2}$, given the mean and standard deviations of the simulated $GD_{p,q}$ distribution. Negative DREX scores indicate that observed GD values are lower (i.e., barcode distributions are more similar) than would be expected by random sampling.

To evaluate the ability of DREX scores to measure the significance of dissemination, we simulated several barcode frequency vectors and varied the extent to which barcodes were shared between samples. First, given a total library size of 1,000 barcodes, we randomly sampled 50 founders twice, representing two independently sampled non-disseminating samples (Fig. 3). As expected, DREX scores between these two unrelated samples were ~0.03 ± 0.92, demonstrating that in the absence of dissemination, GD values are close to those expected by random chance. To establish a threshold for significance of DREX scores, we repeated the simulation of non-disseminating samples 200 times, each for libraries containing 100, 1,000, and 10,000 barcodes, ranging from founding population sizes of 10 to 10,000. Despite the decrease in GD that accompanies higher founding populations, DREX scores were consistently close to 0 (Fig. S1). Therefore, based on these independently sampled, non-disseminating simulations, we set a highly stringent threshold of DREX < −4 ($P = 0.000032$) to infer when sharing of barcodes is significantly greater than would be expected by random chance; obtaining a DREX score less than −4 is extremely unlikely if dissemination is not occurring. The high

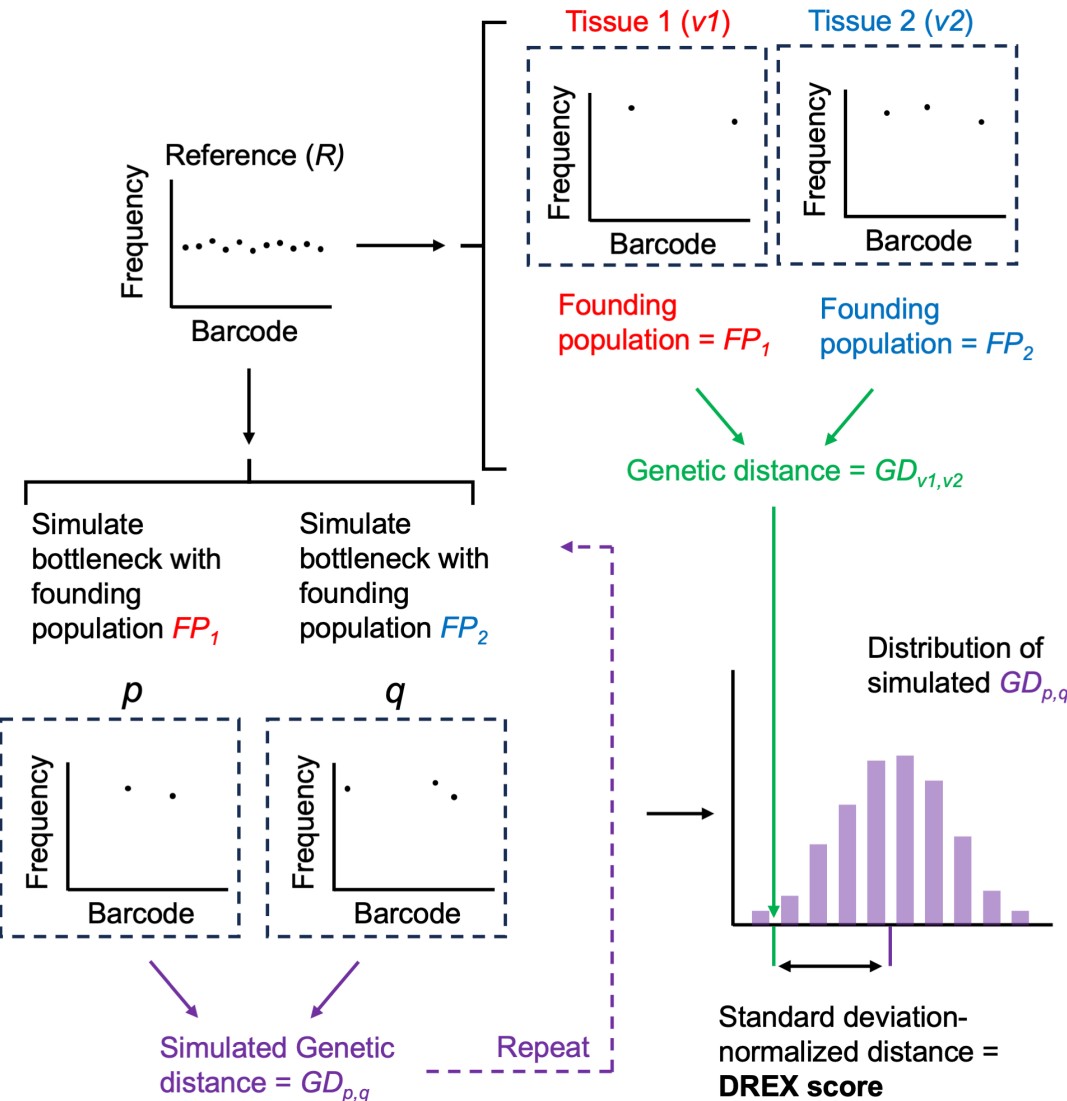

**FIG 2** DREX score calculation. The founding population sizes of two tissue samples ($v_1$ and $v_2$ with founding population sizes $FP_1$ and $FP_2$) are obtained and used to simulate bottlenecks, yielding simulations of non-disseminating samples with identical founding population sizes ($p$ and $q$). This simulation is repeated 50 times, and a distribution of non-disseminating GD values is obtained ($GD_{p,q}$). The observed GD from $v_1$ and $v_2$ is compared against this distribution, where the DREX score is defined as the distance between $GD_{v_1,v_2}$ and the mean of $GD_{p,q}$ simulations, normalized by the standard deviation of $GD_{p,q}$ simulations.

stringency of this threshold is necessary due to the large number of combinatorial comparisons often made in experimental data, such as comparisons between every tissue and every other tissue across biological replicates. Note that for experimental data, DREX score calculation uses the same reference vector (i.e., the barcode distribution in the inoculum) as was used to obtain biological samples. Therefore, the simulations used in DREX score calculation reflect the underlying barcode variability and unevenness, which are unique to each library. In the simulated reference samples above, we assume ~2 logs of variability in barcode abundance, similar to what is observed in (Fig. 3A) practice (6, 9, 23, 24). If future libraries show unexpectedly high levels of variability or unevenness, the reliability of the DREX threshold can be verified by performing similar non-disseminating simulations using these biologically obtained barcoded libraries as the reference. The relationship between GD and DREX scores is schematized in Fig. 3B.

Having established a criterion to derive meaningful dissemination between samples, we next quantified the sensitivity of DREX scores when dissemination is occurring (Fig. 4). Given an initial total library size of 1,000 barcodes, we randomly sampled a population

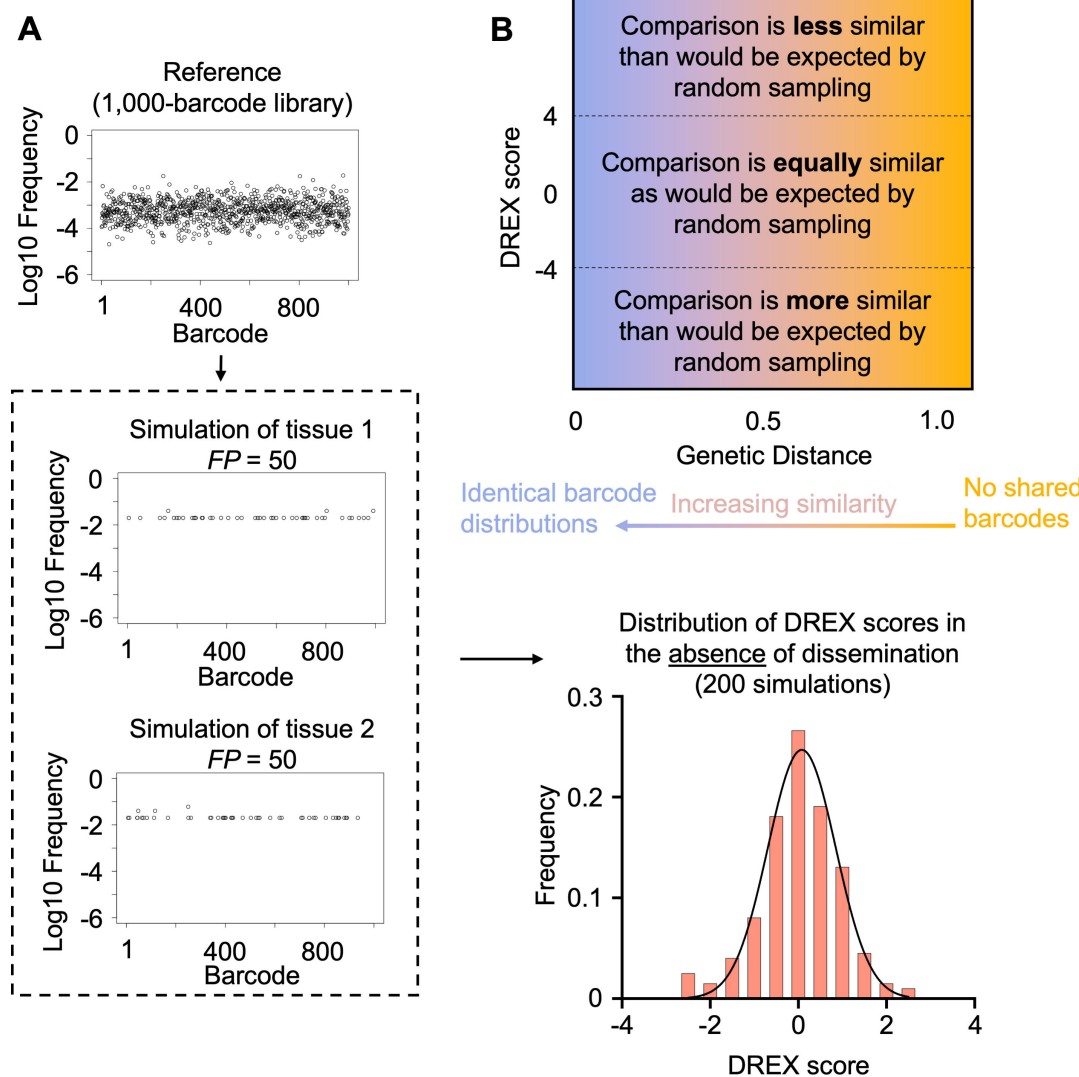

**FIG 3** Distribution of DREX scores in non-disseminating samples. (A) From a simulated library of 1,000 barcodes, 2 samples with founding population sizes of 50 are obtained by multinomial resampling. These simulated samples are input into the STAMPR pipeline to re-calculate founding population sizes (Ns), after which DREX scores are calculated as in Fig. 2. This procedure is repeated 200 times to obtain a distribution of DREX scores representing samples without dissemination. Additional samples with differing library diversities and founding populations are provided in Fig. S1. (B) Conceptual schematic for interpreting GD and DREX scores. At high GDs, samples are dissimilar, while at low GDs, samples are similar. DREX scores between −4 and 4 are as similar as would be expected from random sampling. At low DREX scores, samples are more similar than would be expected by random sampling. At high DREX scores, samples are less similar than would be expected by random sampling.

with a founding population size of 50 twice, representing two independently sampled non-disseminating samples, $v_1$ and $v_2$. We then transferred $k$ barcodes from $v_1$ to $v_2$, simulating the replication of clones in $v_1$ and their subsequent translocation to $v_2$. By increasing $k$, we can define how many barcodes must be transferred, and thus how much GD must decrease, in order for DREX scores to achieve significance. Prior to transferring any barcodes ($k = 0$), $v_1$ and $v_2$ have a GD value of ~0.93 and share approximately six barcodes by random chance, representing ~12% of the total reads in $v_2$. We found that DREX scores reached the −4 threshold when as few as 10 additional barcodes were transferred, representing a GD decrease of ~0.1 and an increase in the proportion of shared barcodes in $v_2$ by 16%. As expected, GD values and DREX scores decreased as $k$ increased. These observations are broadly constant across a range of library sizes and founding population sizes for $v_1$ and $v_2$. Thus, DREX scores are highly sensitive

to genuine dissemination events; with large libraries, a change in GD of as little as 0.03 achieves significance. These simulations also reveal that smaller libraries (e.g., 100 barcodes) require a greater magnitude of dissemination before DREX scores become significant, reflecting the idea that smaller libraries have a greater likelihood of observing identical barcodes across distinct founders (because there are fewer overall barcodes).

In many instances, dissemination between sites is driven by very few clones (very small $k$). To define how abundant these clones must be in order to achieve significant DREX scores, we repeated the simulations of genuine dissemination but set $k$ to 1 (Fig. 5A). To model variable levels of dissemination by a single clone, we increased the relative abundance of the one transferred clone in $v_1$ and $v_2$. For example, when one barcode disseminates in 2 tissues with founding populations of 50 (from a 1,000-barcode library), dissemination is genuine, but the transferred clone is at very low abundance in both samples. Although the total shared barcodes represent ~20% of the reads in both populations, the vast majority of sharing is driven by random sampling of identical barcodes. Correspondingly, DREX scores are not significant (Fig. 5B, no increase in weight/relative abundance). However, increasing the relative abundance (weight) of the single disseminating clone results in smaller DREX scores, which achieve significance when the shared clone decreases GD by 0.06 (increases to ~10% of all reads in both populations). Repeating these simulations across various library and founding population sizes confirms these findings. Notably, in large libraries with samples from small founding populations, a single genuinely disseminating clone that decreases GD by 0.03 (increasing to 7% of all reads) can be detected.

## DREX scores reveal that liver abscess formation drives genuine systemic dissemination

Having established the utility of the DREX score to quantify the significance of dissemination between compartments, we next applied it to previously published barcoding experiments to uncover new facets of within-host dissemination.

We first applied this method to a previous study using a model of *E. coli* systemic infection in mice (30). Here, the authors infected mice intravenously with a ~1,000-barcode library of *E. coli* over a range of inoculum sizes and examined the impact on bottlenecks and dissemination patterns across various tissues. In addition, the authors examined the impact on these dynamics by the LPS sensor TLR4 through the comparison of mice lacking TLR4 (TLR4[KO]) and heterozygous littermates (TLR4[HET]). Of note, a fraction of *E. coli*-infected TLR4[HET] mice form liver abscesses, while TLR4[KO] mice are completely resistant to abscess formation. Abscesses are generally believed to compartmentalize *E. coli*, rather than help facilitate their spread, since GD values (liver/spleen and liver/lung) are similar between TLR4[HET] and TLR4[KO]. However, TLR4[KO] mice have higher founding population sizes than TLR4[HET], and among TLR4[HET] mice, abscess formation is associated with slightly lower GDs. Thus, it remains unclear whether abscess formation leads to genuine dissemination.

We plotted DREX scores against GD values for TLR4[Het] and TLR4[KO] mice, focusing exclusively on comparisons to the liver (i.e., $v_1$ are liver samples, $v_2$ are spleen or lung samples, Fig. 6A). Surprisingly, the mouse genotype itself appeared to clearly segregate by DREX score. Significantly negative DREX scores (less than −4), which indicate that GD values are lower (more similar) than would be expected by random chance, largely consisted of TLR4[HET] mice. Thus, among TLR4[HET] mice, when GD values are lower, the dissemination that occurs between the liver and spleen/lung is likely genuine (Fig. 6A). In contrast, significantly positive DREX scores (more than 4), which indicate that GD values are higher (less similar) than would be expected by random chance, largely consisted of TLR4[KO] mice. Positive DREX scores indicate that the random simulations consistently yielded samples that had greater similarity of barcode distribution than the true biological samples, which is possible if there are differing clonal expansion events in one or both samples that do not disseminate. Tissue-specific clonal expansion was confirmed by examining individual barcode frequency plots (Fig. 6B and C). Critically, the overall GD

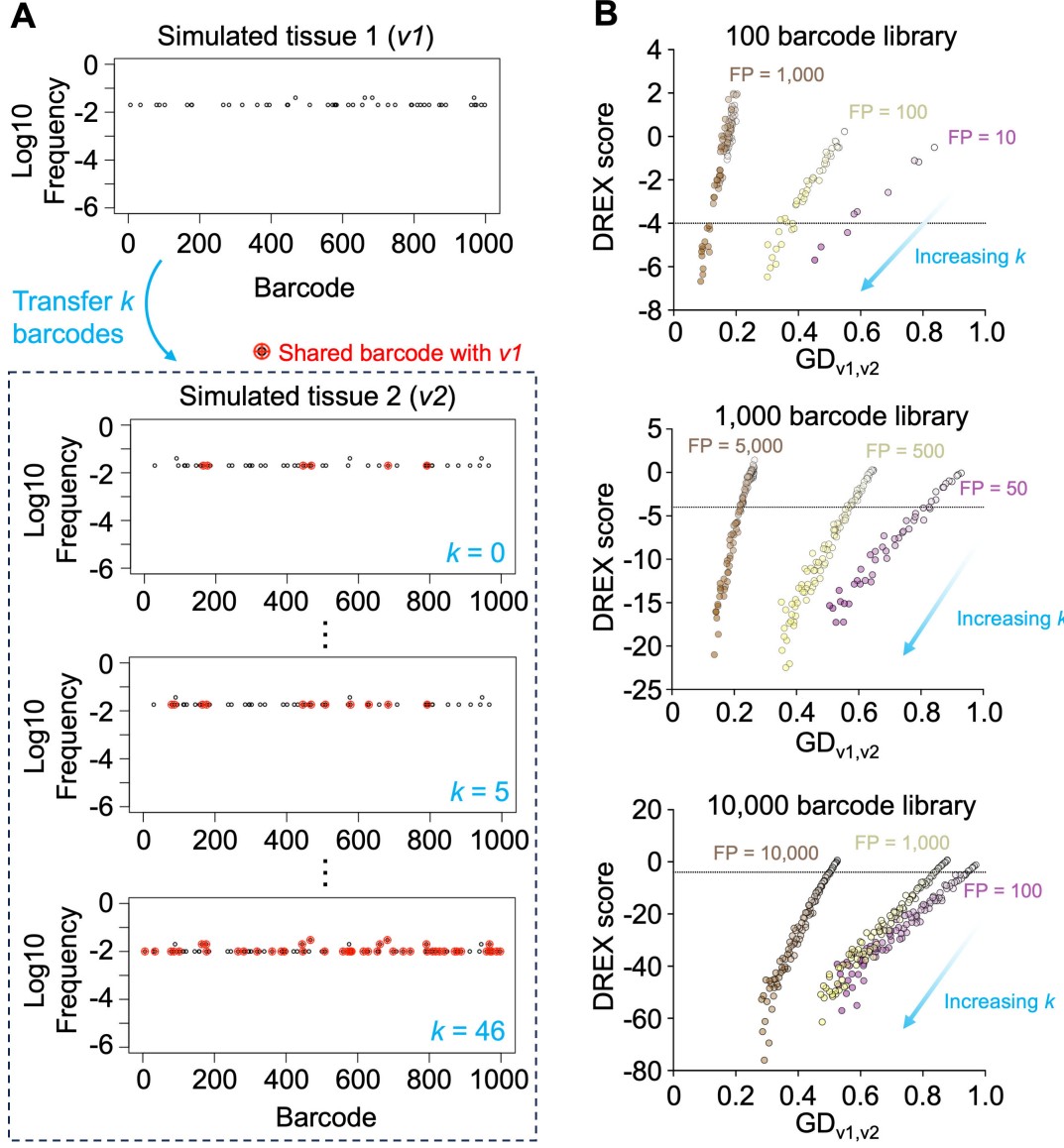

FIG 4  Sensitivity of DREX scores with increasing numbers of transferred clones. (A) Simulations of two tissues, $v_1$ and $v_2$, are generated. Then, $k$ barcodes are transferred from $v_1$ to $v_2$. The value of $k$ is increased (starting from $k = 0$) to simulate a greater number of clones disseminating between the tissues. Barcodes shared when $k = 0$ are those that are shared due to random sampling, and these samples have DREX scores close to 0. (B) Plots of GD versus DREX score when varying library diversity and founding population sizes. As $k$ increases (lighter to darker colors), GD and DREX scores decrease, obtaining significance once enough clones are transferred to exceed the sharing expected by random chance. The sensitivity of DREX scores is interpreted as the change in GD that yields a DREX score of less than $-4$ (defined in Fig. 3; Fig. S1). Note that FP sizes are indicated when $k = 0$, and FP increases with $k$.

values were not significantly different between TLR4$^{Het}$ and TLR4$^{KO}$ (Fig. 6D). Thus, the DREX score provides essential clarification on how dissemination can be interpreted through GD values. Namely, low GD values in TLR4$^{KO}$ mice are not significantly lower (more similar) than would be expected by random chance, and in some cases are higher (less similar) due to differential clonal expansion. In stark contrast, low GD values in TLR4$^{HET}$ mice are driven by genuine dissemination, quantified by low DREX scores.

A notable distinction that could explain the differences in dissemination between TLR4$^{Het}$ and TLR4$^{KO}$ mice is the presence of abscesses almost exclusively in TLR4$^{Het}$ mice. In particular, the observation that TLR4$^{Het}$ but not TLR4$^{KO}$ mice are more likely to possess genuine dissemination may reflect the possibility that liver abscess formation, not the presence of TLR4 *per se*, may drive dissemination to other tissues. To examine this

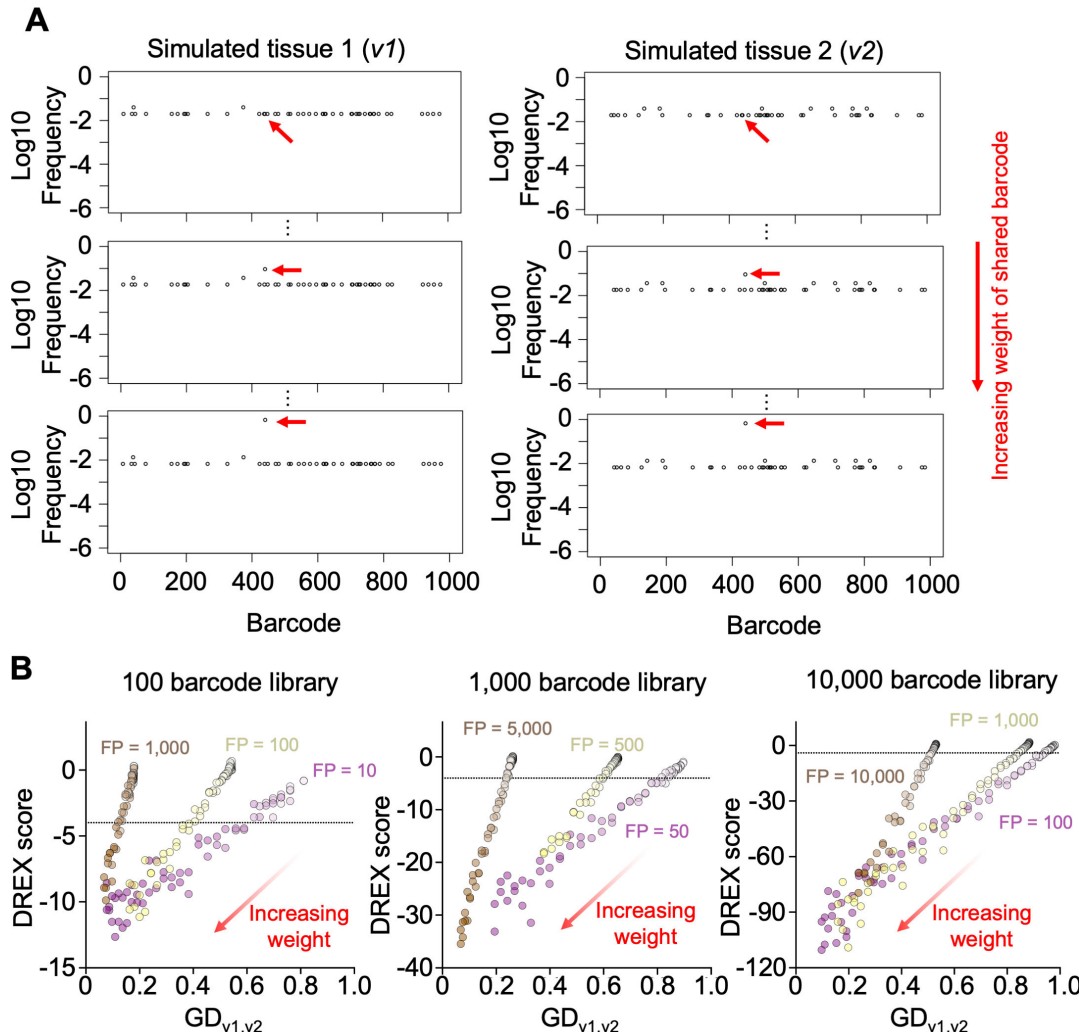

**FIG 5** Sensitivity of DREX scores with increasing abundance of a single shared barcode. (A) Simulations of barcode distributions of two tissues, $v_1$ and $v_2$, are obtained. A single barcode is copied from $v_1$ to $v_2$ (red arrow), and the abundance of this barcode is incrementally increased in both $v_1$ and $v_2$. (B) GD versus DREX scores for various library diversities and founding population sizes using simulations from panel A. As the single barcode increases in abundance (weight), GD and DREX scores decrease. The sensitivity of DREX scores is interpreted as the change in GD that yields a DREX score of less than −4 (defined in Fig. 3; Fig. S1).

possibility, we plotted GD and DREX scores in mice that did and did not form abscesses, irrespective of TLR4 genotype (Fig. 6E). Indeed, mice that developed abscesses had significantly lower and negative DREX scores than mice that did not. Together, these observations indicate that abscess formation in the liver leads to systemic dissemination and underscores how DREX scores can reveal previously hidden patterns of dissemination (Fig. 6F). By lacking abscesses, TLR4$^{KO}$ mice have reduced levels of systemic dissemination. Given the large concentration of immune cells around bacteria within abscesses, we speculate that dissemination from the liver may occur relatively early during abscess development, where *E. coli* has replicated, but immune cells have yet to "wall off" the replicated population. Early dissemination is also consistent with the GD values themselves, which, although they are highly significant by DREX scores, are only modestly low.

## Increase in *Listeria* systemic dissemination in mice lacking Muc2 is not genuine

We next examined dissemination in a model of *Listeria monocytogenes* systemic infection in mice and the influence of the Muc2 mucin, a highly abundant glycoprotein in the intestinal mucus layer (27). Here, the authors orally inoculated a 200-barcode library of

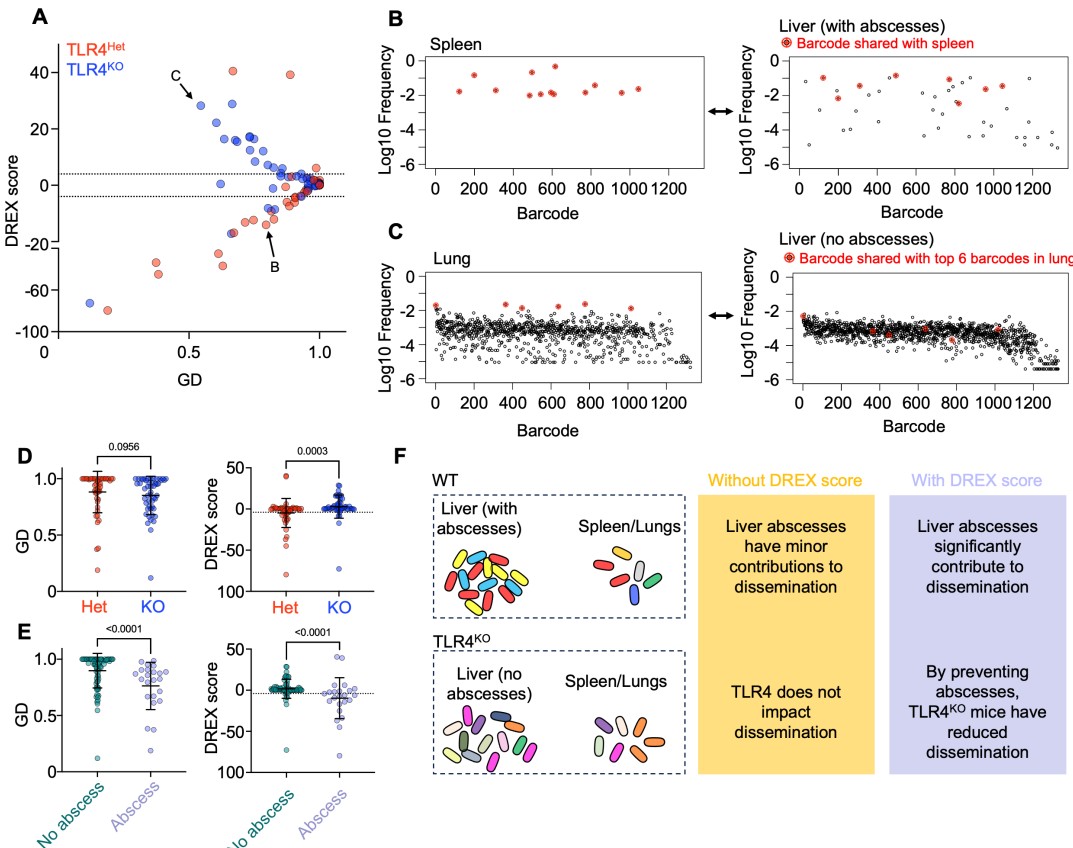

**FIG 6** DREX scores reveal that liver abscesses are associated with dissemination. (A) GD versus DREX scores for TLR4[Het] and TLR4[KO] mice. Tissue comparisons are exclusively those with the liver, relative to the spleen and lungs (35 mice for TLR4[Het] and 34 mice for TLR4[KO]) (B and C) Individual barcode frequency plots from comparisons in panel A. Red barcodes indicate shared clones. (D) GD and DREX scores for TLR4[Het] and TLR4[KO] mice. DREX scores are significantly higher in TLR4[KO] mice and are above 4, indicating that tissue comparisons are less similar than would be expected by random chance. (E) GD and DREX scores separated by mice that developed and did not develop liver abscesses. Not only are GD values significantly lower in mice with abscesses, but these values are also less than would be expected by random chance. (F) Impact of DREX scores on interpretations of dissemination. Abscess formation is associated with significant sharing of clones in wild-type (WT) mice (e.g., the red clone). In the absence of TLR4, abscesses do not form, but founding populations are higher. Therefore, the presence of similar barcodes between the liver and other tissues is more likely due to independent seeding of distinct founders with identical barcodes. Error bars represent means and standard deviations, and *P* values are derived from two-tailed Mann-Whitney tests.

*L. monocytogenes* in mice lacking Muc2 and heterozygous littermates. Note that this study was performed prior to the development of Ns calculation for founding population estimates, and the simulation procedure for DREX scores was modified to account for this distinction (see Materials and Methods). In addition, we focused on only a subset of mice that contained comparisons for all tissues (*n* = 4 for wild type [WT] and *n* = 6 for KO).

Mice lacking Muc2 consistently have larger founding population sizes in both systemic sites and intestinal populations (Fig. 7A). In addition, it was reported that Muc2[KO] mice have lower GD values (increased similarity) between the intestine and systemic sites. To assess whether these lower GD values can be explained by independent seeding of the systemic sites by a greater number of clones, rather than by genuine dissemination from the colon, we plotted DREX scores. We found that despite the decreased GD values in Muc2[KO] mice, DREX scores remained close to 0, indicating that the low GD values are ambiguous and can be explained by independent seeding of different tissues by a greater number of clones (Fig. 7B and C). Critically, these insights must be contextualized with the anatomical route of transit by *L. monocytogenes* following oral inoculation, which must traverse through the intestinal barrier in order to reach systemic sites. Thus, although the absence of Muc2 does lead to an increase in

the number of clones that reach systemic tissues, this insight is inferred from founding population sizes alone; DREX scores demonstrate that there is no evidence from GD values to suggest that the colon and systemic sites share (or do not share) a common pool of replicative bacteria.

Notably, not all dissemination in Muc2$^{KO}$ mice is ambiguous. A hallmark of *L. monocytogenes* infection is bacterial replication in the bile within the gallbladder. Mice lacking Muc2 are more likely to have bacterial replication in bile, and by plotting DREX scores for comparisons with the gallbladder, we detected several instances of significantly low DREX scores, indicating genuine dissemination of other tissues with the gallbladder (Fig. 7C; DREX < −4). These results indicate that Muc2 restricts bacterial entry into the gallbladder, likely by preventing transit out of the intestine, and in turn preventing bacteria from accessing a replicative reservoir for genuine dissemination.

## DREX scores quantify metastatic dissemination in *Klebsiella* infection

DREX scores measure the extent to which GDs between two tissues can be attributed to random sampling, thereby preventing erroneous inferences about which tissues share microbial populations. However, as described in the reanalysis of *L. monocytogenes* experiments, anatomical logic must be considered for such interpretations to ultimately guide the conceptual understanding of dissemination. This idea was encapsulated in a recent study examining *Klebsiella pneumoniae* dissemination from the lung (24). Here, the authors found that mice infected retropharyngeally had high bacterial burdens in the spleen; since the bacteria were inoculated in the lung, they necessarily transited from the lung to the spleen. Barcoding experiments revealed that in some mice, some bacterial clones unevenly expanded in the lungs, and these expanded clones were also present in the spleen. Thus, it was inferred that bacterial replication in the lungs led to dissemination in the spleen, a process termed "metastatic" dissemination. In contrast, in another subset of mice, there was minimal sharing of clones between the lung and spleen, implying the clones that transit from lung to spleen did so before substantial replication in the lung, a process termed "direct" dissemination. Notably, mice exhibiting metastatic or direct dissemination had statistically significant but only minor differences in GD values (Fig. 8A).

Both low DREX scores and metastatic dissemination share the necessary logic that identical barcodes are present in two sites. However, describing dissemination as "metastatic" or "direct" is a binary qualifier, while DREX scores are a quantitative metric that measures the extent to which barcodes are meaningfully shared. Thus, we reasoned that DREX scores would directly quantify the extent to which dissemination is metastatic, in contexts where anatomical logic necessitates a direction of transit (e.g., from lung to spleen). To test this idea, we plotted DREX scores for lung comparisons (versus the spleen, liver, and blood) across mice infected with *K. pneumoniae* at 24 hours post-inoculation (hpi) and binned by whether mice were classified as having metastatic or direct dissemination (defined from lung to spleen in reference 25) (Fig. 8B). We additionally included infected mice lacking NADPH oxidase (encoded by Nox2), which lack the uneven expansion in the lungs and are classified as having direct dissemination to the spleen. Strikingly, all comparisons classified as metastatic exhibited DREX scores less than −4, while most (27/30) comparisons classified as direct had DREX scores greater than this significance threshold. These observations indicate that, when anatomical logic necessitates a direction of microbial transit, DREX scores directly quantify the extent to which replication in the upstream site is required for dissemination.

In addition to conceptually describing metastatic and direct dissemination, the authors also examined how dissemination changes over time by comparing WT mice at 24 and 48 hpi. GD values are lower at 48 hpi than at 24 hpi, suggesting that dissemination increases over time (Fig. 8C). However, as demonstrated extensively above, decreases in GD value can be confounded and must be interpreted by DREX scores. Correspondingly, DREX scores are significantly lower at 48 hpi, indicating that sharing of barcodes is greater than would be expected by random chance (Fig. 8C and D).

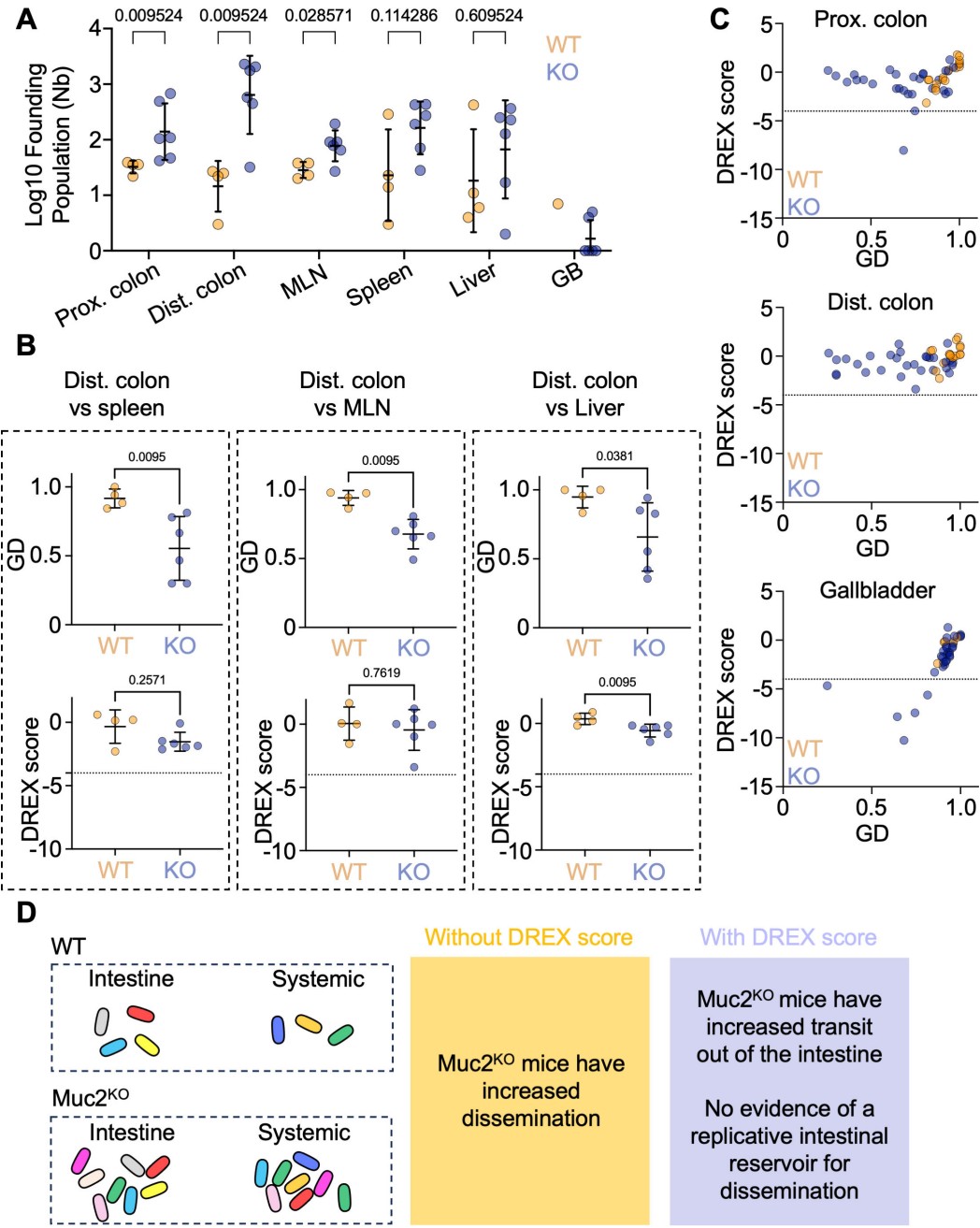

**FIG 7** Founding populations explain changes in GDs during *L. monocytogenes* infection in Muc2[KO] mice. (A) Founding population (as Nb) across the proximal colon (Prox. colon), distal colon (Dist. colon), mesenteric lymph nodes (MLN), spleen, liver, and the bile in the gallbladder (GB) in WT and Muc2[KO] mice (four mice for WT and six mice for Muc2[KO]). Consistently higher founding populations are observed in Muc2[KO] mice. (B) GDs and DREX scores for the spleen, MLN, and liver compared to the distal colon. Muc2[KO] mice have significantly lower GDs but do not achieve significantly low DREX scores. (C) GD versus DREX scores for all comparisons relative to the proximal colon (top), distal colon (middle), and gallbladder (bottom). Despite lower GDs in Muc2[KO] mice, the majority of comparisons do not yield significant DREX scores. Notable exceptions are comparisons to the gallbladder, which represent genuine dissemination events. (D) Impact of DREX scores on interpretations of dissemination. In WT mice, there is very little sharing of barcodes between the intestine and systemic sites. In the absence of Muc2, more barcodes are shared, but this increase in sharing is driven by an increase in founding population sizes rather than genuine dissemination. GD values, therefore, do not provide evidence of a replicative intestinal reservoir for systemic dissemination. Error bars represent means and standard deviations, and *P* values are derived from two-tailed Mann-Whitney tests.

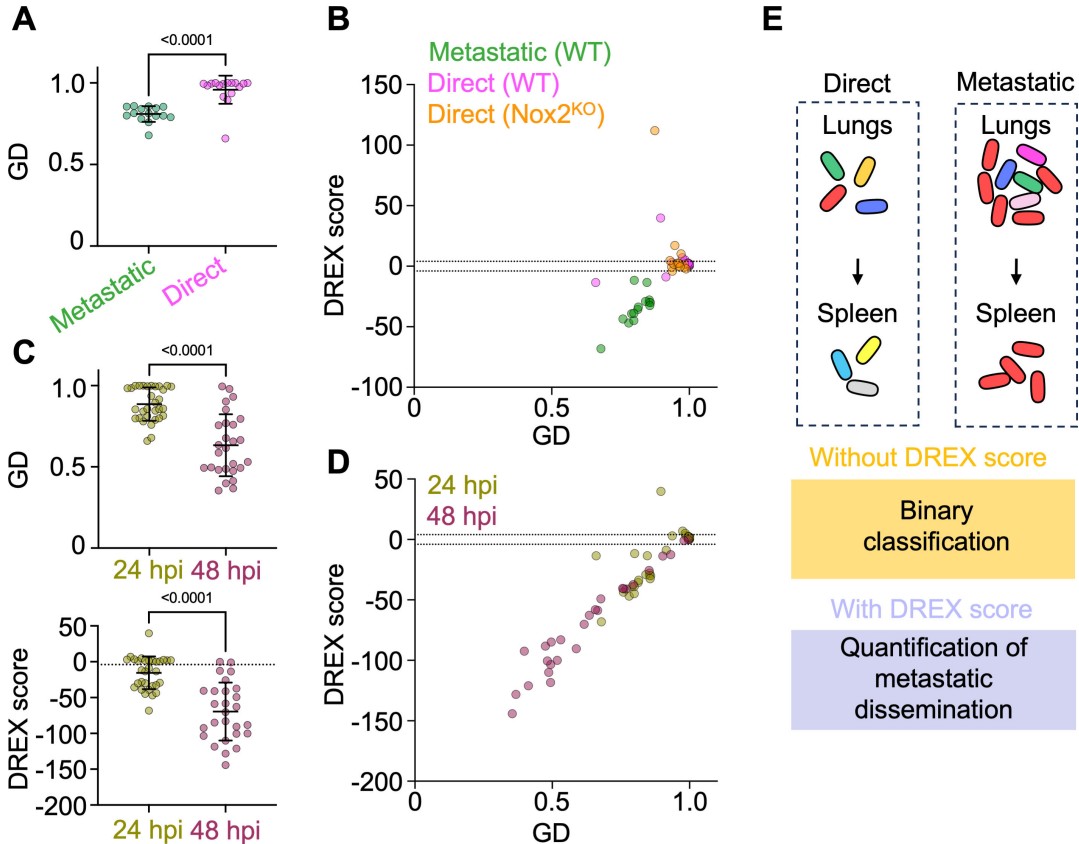

**FIG 8** DREX scores quantify metastatic dissemination in *K. pneumoniae* infection. (A) GD values in WT mice categorized as having metastatic or direct dissemination (from lung/spleen comparisons, six mice with direct dissemination, five mice with metastatic dissemination). Mice with metastatic dissemination have slightly lower GD values. (B) GD versus DREX scores for WT mice with metastatic or direct dissemination and for Nox2$^{KO}$ mice (*n* = 5), which were all categorized as having direct dissemination. DREX scores directly quantify metastatic dissemination. (C) GD and DREX scores for WT mice at 24 and 48 hours post-inoculation (hpi). Lower DREX scores at 48 hpi indicate increased metastatic dissemination over time. Nine mice at 48 hpi. (D) Same data as panel C but plotted together to visualize correlations. (E) Impact of DREX scores on interpretations of dissemination. In mice with direct dissemination, relatively few barcodes are shared between the lungs and spleen. In mice with metastatic dissemination, clonal expansion in the lungs (e.g., red barcode) enables translation to the spleen. This dissemination is genuine, and DREX scores provide direct quantification of the extent to which lung replication is required for systemic spread. Error bars represent means and standard deviations, and *P* values are derived from two-tailed Mann-Whitney tests.

Therefore, since decreases in GD can be confounded, the decrease in significant DREX scores provides a direct and robust quantification to demonstrate how metastatic dissemination increases over time (Fig. 8E).

## Concluding remarks

Here, we demonstrate the utility of the DREX score, representing a simple solution to the problem of high founding population sizes leading to erroneous interpretations of microbial dissemination. This method also enables highly sensitive quantification of the significance of observing shared barcodes between tissues, where even minor changes in GDs can be highly significant. Thus, our metric is particularly well suited for defining circumstances where sub-populations of microbes within a tissue disseminate, exemplified by the *E. coli* clones within liver abscesses. The improved confidence in interpreting dissemination will also be useful in contexts where the overall library diversity is small, as in the *Listeria* reanalysis, where it is important to detect when sharing of barcodes may be due to independent seeding of identical barcodes rather than genuine dissemination. Multiple studies have also reported how dissemination can occur with or without replication in the upstream site, and our method is well suited to measure the extent to which upstream replication is required for dissemination (8,

9). Taken together, we propose that DREX scores should be integrated into the bacterial lineage tracing toolkit as a valuable metric for reliably quantifying dissemination.

## MATERIALS AND METHODS

### Generation of simulated data

RStudio was used for all analyses in this study. To randomly generate barcode count vectors, the following procedures were performed. First, a reference barcode count vector is created containing $n$ elements with the rnorm function (mean = 1, standard deviation = 0.5). This vector is then exponentiated with base 10 and converted to a frequency to create $R$. The $n$ elements in $R$ each represent the frequency of barcodes $b_1$, $b_2$, $b_3$…$b_n$. To artificially bottleneck $R$, thereby creating simulated samples, the rmultinom function is used to sample $R$ FP times (representing various founding population sizes).

Non-disseminating samples are created by repeating the bottleneck simulation 201 times. The first simulation is fixed ($v_1$), and each of the subsequent 200 simulations serves as a comparison sample to the first simulation ($v_{2,1}$, $v_{2,2}$, $v_{2,3}$,…,$v_{2,200}$). This procedure is performed for three different $R$ vectors, each across three values of FP. Since no transfer of barcodes occurs, this procedure represents the null distributions for subsequent calculation of DREX scores.

To simulate dissemination, two distinct approaches are taken. In the first approach, bottleneck simulations are performed twice as above to generate $v_1$ and $v_2$. To model dissemination, $k$ nonzero barcodes in $v_1$ are added to their corresponding counts in $v_2$. The value of $k$ is varied from 0 to the number of nonzero barcodes in $v_1$. The specific barcodes that represent $k$ are chosen using the sample function (without replacement). The number of times $k$ is varied is the minimum value of 100 or the number of nonzero barcodes in $v_1$. This procedure simulates increasing numbers of barcodes transferred from $v_1$ to $v_2$. In the second approach, $k$ is set to 1, but following the transfer of the barcode from $v_1$ to $v_2$, the count of the barcode is increased in both $v_1$ and $v_2$. The abundance of the transferred barcode is incrementally increased from 1.2× to 10,000×. This second approach models a small subpopulation of bacteria representing a single founding clone that replicates and disseminates across two compartments.

### DREX score calculation

The comparison metric used for DREX calculation is a modified Cavalli-Sforza chord distance (GD). GD is calculated from two barcode frequency vectors $v_1$ and $v_2$ as

$$\text{GD}_{v1,v2} = \sqrt{1 - \sum_{i=1}^{n} \sqrt{(b_i, v_1)(b_i, v_2)}},$$

where $b_i, v_1$ represents the frequency of the $i$th barcode in $v_1$, and $b_i, v_2$ represents the frequency of the $i$th barcode in $v_2$. This calculation is different from those in our previous studies, which multiply the resulting value by $(2\sqrt{2})/\pi \sim 0.9003$. We omit this constant so that GD values can range from 0 to 1, rather than 0 to 0.9003. $\text{GD}_{v1,v2}$ is calculated.

DREX score calculation requires five inputs: a reference barcode frequency vector ($R$), two output barcode frequency vectors from biological samples ($v_1$ and $v_2$), and founding population estimates (Ns values) for $v_1$ and $v_2$. The barcode frequency vectors and Ns values are direct outputs from the STAMPR (25) pipeline (19). Note that the DREX calculation is assuming that all nonzero barcodes are truly present in the population, and therefore, the barcode frequency vectors must be corrected for potential noise. Noise correction occurs directly as part of the STAMPR pipeline. DREX scores are the $z$-score normalized distance of $\text{GD}_{v1,v2}$ using the mean and standard deviation of GD values from randomly simulated sample with founding population sizes equivalent to $v_1$ and $v_2$. To obtain these simulated samples, the reference barcode frequency vector is used

for multinomial resampling at founding population sizes of $v_1$ and $v_2$. Each pairwise comparison of biological samples possesses its own unique set of 50 simulations used to calculate the DREX score. For example, if there are 4 tissues collected in a mouse, resulting in 6 unique pairwise comparisons, then 50 unique simulations are performed for each of the 6 pairwise comparisons. In this study, the number of simulations required to obtain DREX scores is set to 50, which is sufficient to achieve stable DREX scores. Performing additional simulations offers diminishing returns and increases the computational time for DREX score calculation (Fig. S2). However, additional simulations can be performed if necessary for increased consistency between runs. Note that by using the same reference vector as is used in biological experiments, these simulation procedures also model the skew that would be expected given the barcode distribution in the inoculum. Ns calculation was performed using the STAMPR pipeline, but all noise correction procedures were omitted for simulated samples, which by definition have no noise (25).

For reanalysis of *Listeria* infection, the input data were modified to account for the fact that, in the original study, the count tables omitted noise correction procedures that are essential for the STAMPR pipeline and subsequent Ns calculation. The original study used Nb to quantify the founding population (not Ns), which estimates founding population sizes based on the relative abundances of each barcode rather than their presence/absence. The distinctions between Nb and Ns are described extensively in reference 25. Since the initial data processing was incompatible with Ns calculation, additional processing was undertaken to ensure that the true number of barcodes was logical, given the founding population values (as Nb). This processing step was performed for an output sample by multinomially resampling the input library Nb times to obtain a new barcode frequency vector. The number of nonzero barcodes in this resampled vector is designated as $j$. The output frequency vector is then ordered from least to greatest, and barcodes are removed (i.e., set to 0 counts) in order until the number of nonzero barcodes in the output vector equals $j$. This procedure is repeated for all samples and ultimately serves to ensure that the random sampling simulations used to generate DREX scores do not use founding population estimates that are substantially less than the number of barcodes detected, which is a known issue with Nb (25). The resulting output samples, therefore, possess fewer barcodes overall, and GD values across most comparisons are increased compared to those reported in Zhang et al. (27). Critically, however, all data reanalyzed with this processing step reproduce the original findings.

## ACKNOWLEDGMENTS

C.L.H. was supported by the National Institute of Allergy and Infectious Diseases of the National Institutes of Health under Award Number R00AI175481. The content is solely the responsibility of the authors and does not necessarily represent the official views of the National Institutes of Health.

## AUTHOR AFFILIATIONS

[1]Department of Microbiology and Immunology, Loyola University Chicago, Maywood, Illinois, USA

[2]Department of Molecular Microbiology, Washington University, St. Louis, Missouri, USA

[3]Department of Microbiology and Immunology, University of Minnesota, Minneapolis, Minnesota, USA

## AUTHOR ORCIDs

Caitlyn L. Holmes  http://orcid.org/0000-0001-5818-148X
Karthik Hullahalli  http://orcid.org/0000-0003-3064-2090

## AUTHOR CONTRIBUTIONS

Rachel T. Giorgio, Formal analysis, Investigation, Methodology, Validation, Visualization, Writing – original draft, Writing – review and editing | My T. Le, Formal analysis, Visualization, Writing – review and editing | Ting Zhang, Data curation, Methodology, Resources, Writing – review and editing | Caitlyn L. Holmes, Data curation, Methodology, Resources, Writing – review and editing | Karthik Hullahalli, Conceptualization, Data curation, Formal analysis, Funding acquisition, Investigation, Methodology, Project administration, Resources, Software, Supervision, Validation, Visualization, Writing – original draft, Writing – review and editing

## DATA AVAILABILITY

The code for DREX score calculation is provided as Code S1. The raw input data for DREX calculation for *E. coli*, *L. monocytogenes*, and *K. pneumoniae* reanalysis are available at 10.5281/zenodo.17573028.

## ADDITIONAL FILES

The following material is available online.

### Supplemental Material

**Code S1 (mSystems01460-25-s0001.rtf).** Code for DREX score calculation.
**Supplemental Figures (mSystems01460-25-s0002.docx).** Figures S1 and S2.

### Open Peer Review

**PEER REVIEW HISTORY (review-history.pdf).** An accounting of the reviewer comments and feedback.

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
