## [Reviewer comments · mSystems]

Accurate interpretation of within-host dissemination using barcoded bacteria

Rachel Giorgio, My Le, Ting Zhang, Caitlyn Holmes, and Karthik Hullahalli

Corresponding Author(s): Karthik Hullahalli, Loyola University Chicago

Review Timeline:

Submission Date:	October 14, 2025
Editorial Decision:	November 9, 2025
Revision Received:	November 26, 2025
Accepted:	December 2, 2025

Editor: Mark Mandel

Reviewer(s): The reviewers have opted to remain anonymous.

Transaction Report:

DOI: <https://doi.org/10.1128/msystems.01460-25>

Re: mSystems01460-25 (**Accurate interpretation of within-host dissemination using barcoded bacteria**)

Dear Dr. Karthik Hullahalli:

You will see that both reviewers were positive on the manuscript overall and offered a number of questions and comments to improve the work. Both reviewers had questions about the number of simulations employed, so please pay special attention to those concerns. Depending on your interpretation of these issues, addressing the comments may require running additional simulations to fully justify the thresholds presented. Aside from potentially those comments, I do not believe that additional experiments are required to address the reviewers' concerns.

Revision Guidelines

Sincerely,
Mark Mandel
Editor
mSystems

Reviewer #1 (Comments for the Author):

This manuscript presents a new metric, the DREX score, that helps to identify and quantify true inter-organ dissemination events

in microbial lineage tracing experiments using libraries of barcoded bacteria. The manuscript is very well-written and clearly demonstrates the DREX score's utility by refining/correcting the interpretation of 3 previously published lineage tracing datasets. I only have very minor comments.

Fig. 1 legend - Define the first use of "GD" (i.e., replace with "genetic distance (GD)") to prevent confusion with "genuine dissemination."

Fig. 1D legend - Change "in B" to "in C."

line 162 - Define the first use of "FP1" (i.e., replace with "founding population size (FP1)") since currently it's only defined in the legend.

line 212 - Some justification for repeating the simulation this specific number of times (50) would be helpful.

line 230 - Some justification for repeating the simulation this specific number of times (200) would be helpful.

line 251 - Missing "to" before "achieve."

Fig. 4 legend - The last sentence is missing a period.

Fig. 5A (right) - Change "v1" to "v2."

Fig. 6F - Missing period.

line 317 - Change "6B" to "6A."

line 326 - Change "6C" to "6BC."

line 343 - Change "6B" to "6F."

Fig. 6F legend (and elsewhere) - Add an additional sentence to help explain and fully make point come across for cartoons. For example, here, the sentence should be something about how the red lineage is locally replicating in abscessed livers prior to true dissemination to distal sites.

Fig. 6D - Missing period.

Fig. 8E - Missing period.

line 444 - Add reference to 8E (currently missing from this section).

Reviewer #2 (Comments for the Author):

In this manuscript, Giorgio et al. present a simulation-based analytical framework that improves interpretation of within-host bacterial dissemination in barcoded infection models. The authors introduce the Distance from Randomly-sampled Expectations (DREX) score, which quantifies whether barcode similarity between tissues exceeds what would be expected by random sampling given the founding population sizes. The approach is carefully implemented, validated with simulated data, and applied to three previously published datasets: *Escherichia coli* liver abscess formation, *Listeria monocytogenes* dissemination in Muc2-deficient mice, and *Klebsiella pneumoniae* pneumonia. The analyses are rigorous, the biological interpretations are meaningful, and the study provides an important corrective to the limitations of genetic distance metrics. Overall, this is an excellent, well-executed study that makes a valuable contribution to microbial lineage tracing. Attention to the following comments would strengthen the manuscript:

Minor Comments

1. The validation simulations use 50-200 iterations per comparison. This may underestimate variance when founding populations are large, where random sampling noise is minimal but still present. The authors should clarify whether increasing iteration depth changes the stability of the DREX distribution, clarify if founding population size or barcode depth alters DREX and recommend how to determine that a number of iterations are sufficient. If computational limits constrain iteration number, this could be stated explicitly.
2. The $DREX < -4$ cutoff is a reasonable and stringent criterion, but its universality is uncertain. The authors should discuss whether this threshold should be recalibrated for each dataset, for example using null distributions from the inoculum library, or whether it is broadly applicable across experimental systems. Along these lines, it would be useful for the author to comment if identification of a DREX cutoff can prospectively inform experimental design including the necessary number of barcodes, founding populations, and/or number of animals needed to determine "true" versus "ambiguous" dissemination.
3. Because DREX relies on founding population estimates from STAMPR, a schematic or brief workflow figure showing how

STAMPR outputs feed into DREX calculations would be useful, particularly for researchers not intimately familiar with the underlying mathematics. Providing an annotated example dataset or simple command-line demonstration in the supplement would further increase accessibility.

4. In the *Listeria* reanalysis, the sample size is limited. The authors should comment on how this affects DREX sensitivity and whether additional replicates would strengthen the conclusions (per comment #2 above, as well.). Across all analyses, please specify the number of biological replicates and pairwise comparisons contributing to each DREX distribution.

5. While "ambiguous dissemination" is defined as a term in the manuscript, it seems to be somewhat of a misnomer in that the finding of barcodes could be either due to dissemination or just dual-colonization by a single barcode population. Therefore, we would suggest reframing this term to be more about ambiguous colonization compared to dissemination since the inherent point made in this manuscript is that we don't know whether this is dissemination versus primary colonization.

6. Include random seeds and software versions in method used for simulations to improve reproducibility.

7. References 8-9 are preprints; update if peer-reviewed versions are available.

8. Correct "linage tracing" to "lineage tracing."

9. Consider noting in the Methods whether DREX scripts will be made publicly available in a repository (GitHub?) for transparency.

Responses in red

Thank you for your positive review of our manuscript. To justify threshold selection, we have added additional Figure S2 to demonstrate the impact of variable simulation number on DREX score, as well as included additional points of discussion. Below is a point-by-point response to reviewer comments:

1. This manuscript presents a new metric, the DREX score, that helps to identify and quantify true inter-organ dissemination events in microbial lineage tracing experiments using libraries of barcoded bacteria. The manuscript is very well-written and clearly demonstrates the DREX score's utility by refining/correcting the interpretation of 3 previously published lineage tracing datasets. I only have very minor comments.

Thank you for your positive comments.

2. Fig. 1 legend - Define the first use of "GD" (i.e., replace with "genetic distance (GD)") to prevent confusion with "genuine dissemination."

Done

3. Fig. 1D legend - Change "in B" to "in C."

Done

4. line 162 - Define the first use of "FP1" (i.e., replace with "founding population size (FP1)") since currently it's only defined in the legend.

Done

5. line 212 - Some justification for repeating the simulation this specific number of times (50) would be helpful.

Justification added in material and methods, along with new Figure S2:

“In this study, the number of simulations required to obtain DREX scores is set to 50, which is sufficient to achieve stable DREX scores. Performing additional simulations offers diminishing returns and increases the computational time for DREX score calculation (Figure S2). However, additional simulations can be performed if necessary for increased consistency between runs.”

6. line 230 - Some justification for repeating the simulation this specific number of times (200) would be helpful.

See response to comment #5

7. line 251 - Missing "to" before "achieve."

Done

8. Fig. 4 legend - The last sentence is missing a period.

Done

9. Fig. 5A (right) - Change "v1" to "v2."

Done

10. Fig. 6F - Missing period.

Done

11. line 317 - Change "6B" to "6A."

Done

12. line 326 - Change "6C" to "6BC."

Done

13. line 343 - Change "6B" to "6F."

Done

14. Fig. 6F legend (and elsewhere) - Add an additional sentence to help explain and fully make point come across for cartoons. For example, here, the sentence should be something about how the red lineage is locally replicating in abscessed livers prior to true dissemination to distal sites.

This is a wonderful suggestion. Added sentences to legends in Figures 6, 7, and 8.

15. Fig. 6D - Missing period.

Done

16. Fig. 8E - Missing period.

Done

17. line 444 - Add reference to 8E (currently missing from this section).

Done

Reviewer #2 (Comments for the Author):

In this manuscript, Giorgio et al. present a simulation-based analytical framework that improves interpretation of within-host bacterial dissemination in barcoded infection models. The authors introduce the Distance from Randomly-sampled Expectations (DREX) score, which quantifies whether barcode similarity between tissues exceeds what would be expected by random sampling given the founding population sizes. The approach is carefully implemented, validated with simulated data, and applied to three previously published datasets: *Escherichia coli* liver abscess formation, *Listeria monocytogenes* dissemination in Muc2-deficient mice, and *Klebsiella pneumoniae* pneumonia. The analyses are rigorous, the biological interpretations are meaningful, and the study provides an important corrective to the limitations of genetic distance metrics. Overall, this is an excellent, well-executed study that makes a valuable contribution to microbial lineage tracing. Attention to the following comments would strengthen the manuscript:

Thank you very much for your positive comments

Minor Comments

1. The validation simulations use 50-200 iterations per comparison. This may underestimate variance when founding populations are large, where random sampling noise is minimal but still present. The authors should clarify whether increasing iteration depth changes the stability of the DREX distribution, clarify if founding population size or barcode depth alters DREX and recommend how to determine that a number of iterations are sufficient. If computational limits constrain iteration number, this could be stated explicitly.

We have added a new Figure S2 to examine the impact of simulation number on DREX score calculation across varying library and founding population sizes:

“In this study, the number of simulations required to obtain DREX scores is set to 50, which is sufficient to achieve stable DREX scores. Performing additional simulations offers diminishing returns and increases the computational time for DREX score calculation (Figure S2). However, additional simulations can be performed if necessary for increased consistency between runs.”

2. The DREX < -4 cutoff is a reasonable and stringent criterion, but its universality is uncertain. The authors should discuss whether this threshold should be recalibrated for each dataset, for example using null distributions from the inoculum library, or whether it is broadly applicable across experimental systems. Along these lines, it would be useful for the author to comment if identification of a DREX cutoff can prospectively inform experimental design including the necessary number of barcodes, founding populations, and/or number of animals needed to determine "true" versus "ambiguous" dissemination.

We have added some additional language to discuss the -4 threshold:

“Note that for experimental data, DREX score calculation uses the same reference vector (i.e., the barcode distribution in the inoculum) as was used to obtain biological samples. Therefore, the simulations used in DREX score calculation reflect the underlying barcode variability and unevenness, which are unique to each library. In the simulated reference samples above, we assume ~2 logs of variability in barcode abundance, similar to what is observed in practice. In future studies, the reliability of the DREX threshold can be verified by performing similar non-disseminating simulations using biologically obtained barcoded libraries as the reference, particularly if libraries show unexpectedly high levels of variability or unevenness.”

Because DREX scores compare a true biologically obtained GD value against several simulations, they are particularly advantageous in defining when biological dissemination is meaningful regardless of sample size, founding population, or barcode diversities. DREX scores do not directly provide prospective insights into experimental design *per se*, but they enable robust conclusions in contexts when some of these parameters may be limiting. For example, some barcoded libraries are difficult to obtain (e.g., poor integration efficiency), resulting in the creation of a relatively low-complexity libraries. Our method enables even these libraries to obtain robust biological insights about dissemination. Similarly, if animal numbers are low, DREX scores enable the dissemination pattern in even a single animal to be meaningful (at least with respect to that specific animal).

3. Because DREX relies on founding population estimates from STAMPR, a schematic or brief workflow figure showing how STAMPR outputs feed into DREX calculations would be useful, particularly for researchers not intimately familiar with the underlying mathematics. Providing an annotated example dataset or simple command-line demonstration in the supplement would further increase accessibility.

We have clarified how the output of the STAMPR pipeline feeds into DREX score calculation in the methods

4. In the *Listeria* reanalysis, the sample size is limited. The authors should comment on how this affects DREX sensitivity and whether additional replicates would strengthen the conclusions (per comment #2 above, as well.). Across all analyses, please specify the number of biological replicates and pairwise comparisons contributing to each DREX distribution.

Added animal number to each figure panel. Added the following to the methods:

“Each pairwise comparison of biological samples possesses its own unique set of 50 simulations used to calculate the DREX score. For example, if there are four tissues collected in a mouse, resulting in six unique pairwise comparisons, then 50 unique simulations are performed for each of the six pairwise comparisons.”

5. While "ambiguous dissemination" is defined as a term in the manuscript, it seems to be somewhat of a misnomer in that the finding of barcodes could be either due to dissemination or just dual-colonization by a single barcode population. Therefore, we would suggest reframing this term to be more about ambiguous colonization compared to dissemination since the inherent point made in this manuscript is that we don't know whether this this dissemination versus primary colonization.

Thank you for this excellent suggestion. Indeed, we have spent an extensive amount of time discussing the terminology in this manuscript. We wish to keep our nomenclature for the following reasons. First, DREX scores are made to quantify the significance of GD values. When investigators are calculating GD values, they are attempting to quantify dissemination between two sites. Thus, keeping the phrase “dissemination” ensures that the terminology is consistent with the underlying goal of the analysis. Second, using the phrase “ambiguous dissemination” parallels well with “genuine dissemination”. Finally, in many cases anatomical logic necessitates that dissemination has occurred (e.g., in *Klebsiella*

infection of the lung). Ultimately, you raise a very important point, which is that the field lacks a clearly defined vocabulary for describing bacterial dissemination. There are clearly several different types of dissemination that vary on how much replication is required in an upstream site, how transient the process is, how synchronous bacterial translocation is, and many other potential factors. We hope that this paper helps facilitate deeper investigation into bacterial dissemination to obtain a more complete lexicon for describing infection dynamics.

6. Include random seeds and software versions in method used for simulations to improve reproducibility.

Random seeds have been added the code and versions are specified in Zenodo, where this data are uploaded.

7. References 8-9 are preprints; update if peer-reviewed versions are available.

Updated

8. Correct "linage tracing" to "lineage tracing."

Done – good catch

9. Consider noting in the Methods whether DREX scripts will be made publicly available in a repository (GitHub?) for transparency.

Added data upload statement

Re: mSystems01460-25R1 (**Accurate interpretation of within-host dissemination using barcoded bacteria**)

Dear Dr. Karthik Hullahalli:

Your manuscript has been accepted, and I am forwarding it to the ASM production staff for publication. Your paper will first be checked to make sure all elements meet the technical requirements. ASM staff will contact you if anything needs to be revised before copyediting and production can begin. Otherwise, you will be notified when your proofs are ready to be viewed.

**** Editor's Note:** The Zenodo DOI in the manuscript submission is not active. Please ensure that the correct URL is listed and that those data are published so that they are publicly-accessible.

Sincerely,
Mark Mandel
Editor
mSystems